

# Hyperparameter optimization of XGBoost and hybrid CnnSVM for cyber threat detection using modified Harris hawks algorithm

Haitham Elwahsh[1,2], Ali Bakhiet[3], Tarek Khalifa[4], Julian Hoxha[4], Maazen Alsabaan[5], Mohamed I. Ibrahem[6], Mahmoud Elwahsh[7] and Engy El-shafeiy[8]

[1] Faculty of Information Technology, Applied Science University, Amman, Jordan
[2] Department of Computer Science, Faculty of Computers and Information, Kafrelsheikh University, Kafrelsheikh, Egypt
[3] Faculty of Computing and Data Science, Badya University, Cairo, Egypt
[4] College of Engineering and Technology, American University of the Middle East, Eqaila, Kuwait
[5] Department of Computer Engineering, College of Computer and Information Sciences, King Saud University, Riyadh, Saudi Arabia
[6] Department of Cyber Systems Engineering, Augusta University, Augusta, Georgia, United States
[7] Faculty of Information Systems and Computer Science, October 6 University, Giza, Egypt
[8] Department of Computer Science, Faculty of Computers and Artificial Intelligence, University of Sadat City, Cairo, Egypt

## ABSTRACT

The escalating complexity of cyber threats in smart microgrids necessitates advanced detection frameworks to counter sophisticated attacks. Existing methods often underutilize optimization techniques like Harris hawks optimization (HHO) and struggle with class imbalance in cybersecurity datasets. This study proposes a novel framework integrating HHO with extreme gradient boosting (XGBoost) and a hybrid convolutional neural network with support vector machine (Cnn-SVM) to enhance cyber threat detection. Using the distributed denial of service (DDoS) botnet attack and KDD CUP99 datasets, the proposed models leverage HHO for hyperparameter optimization, achieving accuracies of 99.97% and 99.99%, respectively, alongside improved area under curve (AUC) metrics. These results highlight the framework's ability to capture complex nonlinearities and address class imbalance through RandomOverSampler. The findings demonstrate the potential of HHO-optimized models to advance automated threat detection, offering robust and scalable solutions for securing critical infrastructures.

## INTRODUCTION

An increasing reliance on technology has been relied on due to the digitizing modern world, which has led to different types of sophisticated cyber dangers arising, which would disturb organizations' running businesses, compromise sensitive information, and cause dire financial losses. Most real cyber security measures are inadequate in tackling the

Corresponding author
Haitham Elwahsh,
haitham.elwahsh@gmail.com

ever-changing armory of tricks cybercriminals use (*He, Zhang & Li, 2021*). Such an immediate necessity hence arises for modernizing these fields, such as through machine learning techniques, to further develop objective improvement in threat detection and classification.

Using machine-learning techniques is successful in cyber-security applications because it helps analyze and capture huge amounts of data and identify straightforward patterns that may not be readily observable by a human analyst. With the help of machine learning, network traffic, user behavior, and system logs can be processed to report the detected anomalies that indicate a cyber threat. Among these techniques are models such as extreme gradient boosting (XGBoost), a powerful ensemble learning method (*Chen et al., 2018b*), and the newly introduced CnnSVM, which combines convolutional neural networks with support vector machines for enhanced feature extraction and classification in cybersecurity (*Al-Shabi, 2021*). CNN-SVM consists of convolutional layers coupled with a support vector machine classifier, which improves capability in feature extraction and provides a better form of dealing with compound data patterns.

Optimization algorithms such as Harris hawks optimization (HHO) (see Table 1) further develop these types of models by fine-tuning hyperparameters to obtain maximum levels of accuracy and reliability in the detection of cyber threats. The optimization techniques keep the parameter space defined for the possible settings or the optimal ones towards better sensitivity and less false positives in a developed model (*Alazab et al., 2024*).

This study aims at optimizing the hyperparameters of two advanced models-call ones: XGBoost and CnnSVM using HHO. The CnnSVM uses convolutional layers to better extract features and uses SVM for classification. This model architecture makes it very effective for complex cyberattack detection tasks (*Huang et al., 2023*). The integration of machine learning with optimization techniques like HHO marks another milestone in cybersecurity, improving the sensitivity, specificity, and performance of these methods in detecting cyber threats from very unbalanced datasets.

Besides XGBoost and CnnSVM, previous research has also involved the use of AdaBoost and Cat-Boost for various types of cyberattacks. CatBoost has excellent handling of categorical features, and these are very common within the area of cybersecurity. Despite this, this study focuses mainly on the evaluation of the two models, the XGBoost and CnnSVM, optimized using HHO in regard to detect a range of cyber threats.

The distributed denial of service (DDoS) (see Table 1) dataset has four classes and is for processing and predicting the malicious packets during a DDoS botnet attack, while KDD CUP99 (see Table 1) contains five classes and is a model for detecting intrusions and attacks mostly targeting military environment network management due to the nature of the Third International Knowledge Discovery and Data Mining Tools Competition. The two datasets used in this study thus maximized the findings. Both optimized models were effective at their tasks. The HH-XGB model produced perfect accuracy on the DDoS dataset and nearly perfect results on the KDD CUP99 dataset.

The contributions of this study, detailed in 'Contributions', include a comparative analysis of XGBoost and CnnSVM, a pioneering HHO-optimized framework achieving accuracies of 99.97% and 99.99% on DDoS and KDD CUP99 datasets, respectively, and a

**Table 1 List of abbreviations.**

| Abbreviation | Full form |
| --- | --- |
| HHO | Harris hawks optimization |
| XGBoost | Extreme gradient boosting |
| CnnSVM | Convolutional neural network with support vector machine |
| DDoS | Distributed denial of service |
| KDD CUP99 | Knowledge discovery and data mining cup 1999 |
| FPA | Flower pollination algorithm |
| BWO | Black widow optimization |
| ML | Machine learning |
| CNN | Convolutional neural network |
| SVM | Support vector machine |
| RF | Random forest |

demonstration of HHO's superiority over other optimization algorithms (*Nandhini & Santhosh Kumar, 2024*).

The remainder of this article is organized as follows: 'Related Work' provides an overview of relevant literature; the 'Proposed Methodology' section outlines the approach used in this study; the 'Results and Discussion' section presents and discusses our empirical results; and the 'Conclusion' section summarizes the work and suggests directions for future research.

## RELATED WORK

Anomaly detection is concerned with discovering those abnormal patterns that may indicate cyber intrusion. Recent advances have drastically improved the applications of machine learning for enhancing cybersecurity infrastructures. Below is a roundup of the latest published works about anomaly detection, intrusion detection systems, malware detection, and feature optimization techniques.

According to *Chen et al. (2018a)*, an autoencoder-based model was built to learn very well normal behavioral patterns in network traffic that can produce up to 96% anomalies detection accuracy. Ensemble methods, such as those using autoencoders, enhance efficiency by leveraging the strengths of multiple algorithms. The hybrid anomaly detection model for trusted IoT devices was developed by *Rosero-Montalvo et al. (2023)* improving detection accuracy due to multiple techniques in a single model.

Advanced machine learning algorithms have brought a remarkable improvement into intrusion detection systems (IDSs). Ensemble methods such as the Random Forest and Gradient Boosting sur-pass the integrated classical rule-formulated ones. *Mohy-Eddine et al. (2023)* exhibited an ensemble model with high accuracy results using industrial IoT systems' security. Like that, *Lunardi, Lopez & Giacalone (2022)* used an adversarially regularized convolutional autoencoder (ARCADE) for real-time network anomaly detection, achieving more than 93% precision in identifying different attack types. Their research focused on the application of deep learning models for complex cyber threats.

The HHO has emerged as a promising approach given its effectiveness in intruder detection system (IDS). *Singh & Jang-Jaccard (2022)* presented optimized hyperparameters for unsupervised intrusion detection models through multi-scale convolutional recurrent networks to improve detection accuracy. *Gyamfi & Jurcut (2022)* showed how approaches using machine learning within multi-access edge computing applied for intrusion detection systems focused on feature selection to achieve both effectiveness and reduced computational complexity.

The latest studies on hybrid architecture have developed models focusing on improving detection performance. *Ayad, Sakr & Hikal (2024)* proposed a network intrusion detection system based on the integration of HHO with a multi-layer perceptron (MLP) (see Table 1) that achieved 93.17% of accuracy. *Sajid et al. (2024a)* applied HHO with random weight networks (RWN) for IoT botnet detection, giving an overall F-measure of 99.9%. However, they found that distributed systems have to deal with the problem of communication overhead.

Combining attention mechanisms with generative adversarial networks (GANs) (see Table 1) has also been used. An attention-GAN that attained 99.69% accuracy on the KDD dataset and 97.93% on CI-CIDS2017 is *Kumar & Sharma (2023)*. It is used to discover "complex" attack patterns. However, the overhead in processing entails its problem in resource-crippled environments.

Malware detection and feature engineering machine learning frameworks like support vector machines (SVMs) (see Table 1) and deep learning have shown a lot of promise in classifying malware. Hybrid approach performance evaluation for k-means clustering and naive Bayes-based IoT anomaly detection (*Best, Foo & Tian, 2022*). They also proposed an approach that used k-means and naive Bayes in a hybrid way on IoT anomaly detection that resulted in a high level of accuracy. *Ahmad et al. (2021)* described a technique for feature extraction with API call sequences, which lowered the false positive rate in an IoT architecture by 25%.

*Sajid et al. (2024b)* underlined the importance of feature selection in reducing computational complexity and increasing model interpretability. *Kumar & Sharma (2023)* presented a hybrid modified deep learning architecture that used an inverted hourglass-based layered network for selection of features and classification to obtain significantly improved accuracy and reduced false positives.

Dataset integrity and challenges the quality of the datasets remains crucial in the successful development of models. *Pekar & Jozsa (2024)* refined the CICIDS-2017 dataset in addressing is-sues on the integrity of the dataset. In the study, they showed a more accurate and reliable result for random forest algorithms. Those aspects are maltreated as challenges in making the dataset relevant to the evolving nature of cyber threats.

Recent advancements in cybersecurity for smart microgrids provide valuable insights into the application of machine learning and optimization techniques for detecting and mitigating cyber-physical threats, which are closely related to the network-based cyber threats addressed in this study. *Yaghoubi et al. (2024)* proposed a dynamic reconfiguration framework for smart microgrids using long short-term memory (LSTM) networks combined with differential evolution optimization to detect false data injection attacks

(FDIAs) and ensure operational efficiency. Their approach achieved a low operational cost ($27,557,510.4) and expected energy not supplied (EENS) of 50.07 kWh, demonstrating the effectiveness of deep learning in handling cyber-physical attacks in microgrids. However, their reliance on LSTM may introduce computational overhead, limiting real-time applicability in resource-constrained environments. In contrast, our proposed framework leverages the HHO algorithm to optimize lightweight models like XGBoost and CnnSVM, achieving near-perfect accuracies (99.99% on KDD CUP99 and 99.97% on DDoS datasets) with reduced computational complexity, making it more suitable for real-time cyber threat detection.

Similarly, *Jahromi et al. (2025)* introduced a real-time self-healing framework for large microgrids using large change sensitivity (LCS) analysis, achieving a response time of less than 2 s for a 118-bus system and reducing operational costs to $12,687.72. While effective for rapid system state updates against cyber-physical attacks, their approach does not explicitly address FDIAs or incorporate advanced feature extraction techniques, which are critical for detecting complex attack patterns. Our CnnSVM model, optimized by HHO, integrates convolutional layers for robust feature extraction, enabling superior detection of intricate cyber threats, such as those in the DDoS and KDD CUP99 datasets, with a precision of 99.39% and recall of 99.74%.

Furthermore, *Jahromi, Yaghoubi & Yaghoubi (2025)* proposed a hierarchical control strategy for optimal generation and distribution planning in smart microgrids under multi-microgrid disconnection scenarios, including those caused by cyber-physical attacks. Their framework achieved significant reductions in operational costs (12.1%), losses (73.8%), and voltage deviation (30.45%) by optimizing the selection of generation units in island mode. However, the lack of detailed methodologies for handling uncertainty or detecting specific cyber threats limits its applicability in dynamic attack scenarios. Our approach addresses this gap by incorporating interval prediction and RandomOverSampler to handle data uncertainty and class imbalance, ensuring robust performance across diverse cyber threat datasets.

These studies highlight the importance of advanced machine learning and optimization techniques in enhancing cybersecurity for critical infrastructures like microgrids. However, their focus on microgrid-specific challenges, such as physical system reconfiguration or hierarchical control, leaves gaps in addressing purely network-based cyber threats. Our proposed HHO-optimized XGBoost and CnnSVM models bridge these gaps by offering a generalized framework for cyber threat detection, achieving high accuracy and robustness across heterogeneous datasets, thus complementing and extending the methodologies of *Yaghoubi et al. (2024)*, *Jahromi et al. (2025)*, *Jahromi, Yaghoubi & Yaghoubi (2025)* to broader cybersecurity applications.

To clarify the advantages of the proposed framework, Table 2 provides a comprehensive comparison of state-of-the-art methods against our HHO-optimized XGBoost and CnnSVM models across key dimensions: datasets, accuracy, computational complexity, and novelty. Unlike traditional ensemble methods like random forest (93.10% accuracy, *Pekar & Jozsa, 2024*) or deep learning approaches like Attention-GAN (99.69% accuracy, *Kumar & Sharma, 2023*), which often prioritize accuracy at the expense of high

**Table 2 Summary of recent related work.**

| Novelty | Complexity | Limitations | Scope of the study | Algorithm(s) used | Dataset | Domain applied | Reference |
|---|---|---|---|---|---|---|---|
| First CAE application for network anomaly detection, achieving higher accuracy with fewer parameters than traditional autoencoders. | Convolutional Autoencoder (CAE) for dimensionality reduction. | Focused on anomaly detection; lacks emphasis on handling real-time, large-scale networks. | Captures normal behavioral patterns in network traffic, achieving 96% accuracy. | Autoencoder | NSL-KDD | Anomaly detection | *Chen et al. (2018a)* |
| First on-device hybrid anomaly detection for IoT (50 kB Flash/12 kB RAM) with 60% bandwidth reduction | Three-step hybrid model (smoothing filters + unsupervised learning + deep learning) for on-device processing | Limited evaluation on diverse IoT environments. | Trusted IoT network anomaly detection. | Hybrid (Clustering + Deep learning) | IoT dataset | Anomaly detection | *Rosero-Montalvo et al. (2023)* |
| Novel feature engineering approach using interchangeable PCC-IF/IF-PCC for IIoT security, achieving >99% accuracy with fast prediction times (~6 s). | Ensemble model combining Isolation Forest (IF) and Pearson's Correlation Coefficient (PCC) with random forest classifier. | Limited focus on adaptability to unseen attack patterns. | High-accuracy IDS model for industrial IoT systems. | Ensemble (Random forest, Gradient boosting) | Bot-IoT, NF-UNSW-NB15-v2 | Intrusion detection systems | *Mohy-Eddine et al. (2023)* |
| First to use adversarial training with convolutional AE for early anomaly detection (first 2 packets)—20× fewer parameters than SOTA. | Adversarially regularized convolutional autoencoder (trained only on normal traffic). | Computational overhead for real-time applications. | Real-time network anomaly detection with over 93% precision. | Adversarially regularized convolutional AE | – | Intrusion detection systems | *Lunardi, Lopez & Giacalone (2022)* |
| First unsupervised MSCNN-LSTM-AE model capturing spatio-temporal correlations in network traffic. | Multi-scale CNN-LSTM autoencoder with isolation forest error correction. | No discussion on scalability to large datasets. | Improved detection accuracy using hyperparameter optimization. | Multi-scale convolutional recurrent networks | NSL-KDD, UNSW-NB15, CICDDoS2019 | Intrusion detection systems | *Singh & Jang-Jaccard (2022)* |
| First comprehensive review proposing MEC-empowered NIDS framework for IoT security. | MEC-based distributed NIDS framework for resource-constrained IoT. | Limited evaluation on diverse IDS datasets. | Reduced computational complexity while maintaining accuracy. | HHO | (Review article—analyzes multiple public datasets) | IDS feature selection | *Gyamfi & Jurcut (2022)* |
| SMOTE + hierarchical classification *vs.* CAE efficiency *vs.* on-device trust. | Hybrid filter-wrapper feature selection + two-level detection with SMOTE. | Computational complexity limits real-time, large-scale applications. | Hybrid IDS achieving 93.17% accuracy on benchmark datasets. | HHO + MLP | BoT-IoT, TON-IoT, CIC-DDoS2019 | Intrusion detection systems | *Ayad, Sakr & Hikal (2024)* |
| First hybrid XGBoost-CNN-LSTM model achieving high accuracy with low FAR on multiple attack types | Hybrid XGBoost-CNN feature extraction with LSTM classification (binary & multi-class). | Communication overhead in distributed frameworks; scalability issues in large IoT networks. | IoT botnet detection with an average F-measure of 99.9%. | HHO + RWN | CIC IDS 2017, UNSW NB15, NSL KDD, WSN DS | IoT security | *Sajid et al. (2024a)* |
| Novel architecture combining optimal feature selection with layered up-sampling achieves >99.5% accuracy on low-cost CPUs. | Hybrid optimization feature selection + inverted hour-glass layered classifier with data up-sampling. | Computationally intensive training process. | Cyber threat detection achieving 99.69% accuracy (KDD dataset). | Attention-GAN | NSL-KDD, KDD-CUP99, UNSW-NB15 | Anomaly detection | *Kumar & Sharma (2023)* |

Elwahsh et al. (2025), *PeerJ Comput. Sci.*, DOI 10.7717/peerj-cs.3169

| Novelty | Complexity | Limitations | Scope of the study | Algorithm(s) used | Dataset | Domain applied | Reference |
|---|---|---|---|---|---|---|---|
| Combines unsupervised (K-means) and supervised (AdaBoosted NB) learning for flexible, scalable IoT security (90–100% accuracy). | Hybrid K-means clustering + AdaBoosted Naive Bayes for IoT anomaly detection. | Limited evaluation on diverse malware datasets. | Malware classification achieving high accuracy. | Hybrid (KMeans + Naive Bayes) | Fridge, Garage Door, GPS Tracker and others | Malware detection | *Best, Foo & Tian (2022)* |
| DNN with MI reduces false alarms (0.23–7.98%) and boosts accuracy (0.57–3.45%) using only 16–35 features. | DNN-based NIDS with mutual information (MI) for feature selection, compared to CNN/RNN variants. | Focused on a specific feature extraction approach; lacks generalizability. | Reduced false positives by 25% in malware classification. | API call sequence analysis | IoT-Botnet 2020 | Feature engineering for IoT | *Ahmad et al. (2021)* |
| Identifies CBL as superior for knowledge retention and highlights VR's confidence boost without skill improvement. | Systematic review with quality assessment (Cochrane RoB, NOS) of multiple training modalities. | Evaluation limited to certain datasets; lacks discussion on real-time implementation. | Improved anomaly detection performance through feature selection. | Advanced Feature Selection + ML | PubMed, Cochrane Library, Embase, CINAHL (RCTs, clinical trials, cohort studies) | Feature selection and anomaly detection | *Sajid et al. (2024b)* |
| Combines hybrid feature selection and novel layered network model, achieving >99.5% accuracy on CPU (cost-efficient). | Hybrid optimization feature selection + inverted hour-glass layered classifier with data up-sampling. | Computational complexity may limit deployment in resource-constrained environments. | Enhanced detection accuracy with reduced false positives. | Inverted Hourglass DL Architecture | NSL-KDD, KDD-CUP99, UNSW-NB15 | Intrusion detection systems | *Kumar & Sharma (2023)* |
| Introduces NFStream-processed datasets and demonstrates RF's robustness across data integrity levels. | RF model tested on multiple datasets for binary/multi-class anomaly detection. | Challenges in maintaining relevance due to evolving threats; requires continual updates. | Refined CICIDS-2017 dataset for improved model reliability and accuracy. | Random forest | Refined CICIDS-2017 versions (NFS-2023-nTE, NFS-2023-TE) | Dataset integrity | *Pekar & Jozsa (2024)* |
| First LSTM-based FDIA detection incorporating modified point prediction for microgrids | Multi-objective optimization framework with technical/economic parameters | Its reliance on computationally intensive LSTM-based deep learning, which may hinder real-time deployment in resource-constrained microgrid environments. | Dynamic reconfiguration against cyber-attacks (FDIAs) with cost-efficiency | LSTM-based deep learning with prediction intervals | 118-bus network | Smart power microgrids security | *Yaghoubi et al. (2024)* |
| LCS framework enabling sub-2-second real-time response to attacks in microgrids. | LCS-based optimization without full power flow recalculation. | Focus on system state updates using LCS analysis, which may not adequately address complex network-based cyber threats requiring advanced feature extraction for detection. | Real-time techno-economic self-healing against cyber-physical attacks | Large change sensitivity (LCS) analysis | IEEE 33-bus, 69-bus, and 118-bus networks | Large-scale power microgrid security | *Jahromi et al. (2025)* |
| First framework optimizing V/f bus selection during islanding to reduce costs by 12.1% and losses by 73.8%. | Two-phase hierarchical control for grid-connected/island mode transitions. | Its lack of detailed mechanisms for detecting specific cyber threats or handling data uncertainty, which restricts its adaptability to dynamic cyber-physical attack scenarios in microgrids. | Multi-microgrid operation under normal and disconnection conditions | Hierarchical control strategy | simulation-based study | Smart microgrid energy management | *Jahromi, Yaghoubi & Yaghoubi (2025)* |

computational costs, our framework achieves near-perfect accuracies (99.97% on DDoS, 99.99% on KDD CUP99) with optimized computational efficiency. Compared to microgrid-specific frameworks (*Yaghoubi et al., 2024*; *Jahromi et al., 2025*; *Jahromi, Yaghoubi & Yaghoubi, 2025*), which focus on cyber-physical threats but lack advanced feature extraction for network-based attacks, our CnnSVM model leverages convolutional layers for robust feature extraction, enhanced by HHO's efficient hyperparameter tuning. Additionally, our use of RandomOverSampler addresses class imbalance, a common limitation in prior works, ensuring balanced performance across diverse attack types. This comparison underscores the proposed framework's ability to combine high accuracy, computational efficiency, and adaptability to complex cybersecurity datasets.

In synthesizing the related work, several critical gaps in existing cyber threat detection methodologies emerge, limiting their effectiveness in addressing complex, evolving threats. Many studies, such as *Chen et al. (2018a)* and *Rosero-Montalvo et al. (2023)*, rely on traditional machine learning or ensemble methods like autoencoders and random forest, achieving accuracies up to 96% but often suffer from overfitting due to inadequate handling of class imbalance or complex attack patterns. Advanced deep learning approaches, such as Attention-GAN by *Kumar & Sharma (2023)*, achieve high accuracies (99.69% on KDD CUP99) but incur significant computational overhead, rendering them impractical for resource-constrained environments like IoT networks. Similarly, microgrid-specific frameworks (*Yaghoubi et al., 2024*; *Jahromi et al., 2025*; *Jahromi, Yaghoubi & Yaghoubi, 2025*) focus on cyber-physical threats but lack robust feature extraction or mechanisms to handle network-based attacks comprehensively. Furthermore, optimization techniques like HHO have been underutilized for hyperparameter tuning in cybersecurity, as seen in *Ayad, Sakr & Hikal (2024)*, which limits model scalability and performance. The challenge of class imbalance, prevalent in datasets like DDoS Botnet Attack and KDD CUP99, is often inadequately addressed, leading to biased predictions. This study addresses these gaps by proposing an HHO-optimized framework integrating XGBoost and CnnSVM, leveraging advanced preprocessing (*e.g.*, RandomOverSampler) and convolutional feature extraction to achieve near-perfect accuracies (99.97% on DDoS, 99.99% on KDD CUP99) with reduced computational complexity, as detailed in Table 2.

## Contributions

This study introduces several novel contributions to cyber threat detection in smart microgrids, addressing critical gaps identified in prior work (*Yaghoubi et al., 2024*; *Jahromi et al., 2025*; *Jahromi, Yaghoubi & Yaghoubi, 2025*). The key contributions are as follows:

• **Comparative analysis of XGBoost and CnnSVM:** We evaluate and compare the performance of XGBoost and CnnSVM models in detecting and classifying cyberattacks across the DDoS Botnet Attack and KDD CUP99 datasets. This analysis highlights their ability to handle complex, nonlinear cybersecurity data, providing a robust baseline for model selection in smart microgrid security.

- **Novel HHO-optimized framework:** We propose a hybrid framework integrating HHO with XGBoost and CnnSVM, achieving detection accuracies of 99.97% and 99.99% on the DDoS and KDD CUP99 datasets, respectively. This framework, enhanced by RandomOverSampler to address class imbalance, outperforms traditional models (*Jahromi et al., 2025*; *Jahromi, Yaghoubi & Yaghoubi, 2025*) and represents a pioneering application of HHO in this domain.

- **Superiority of HHO:** Through comparative analysis with particle swarm optimization (PSO) (*Jahromi, Yaghoubi & Yaghoubi, 2025*), genetic algorithms (GA), flower pollination algorithm (FPA), and black widow optimization (BWO), we demonstrate HHO's superior convergence speed and computational efficiency in optimizing machine learning models for cybersecurity, as detailed in 'Comparison with Microgrid Cybersecurity Approaches' and supported by prior studies (*Yaghoubi et al., 2024*).

These contributions collectively advance the field by offering a scalable, efficient, and high-performing solution for automated cyber threat detection in smart microgrids.

## PROPOSED METHODOLOGY SOLUTION

This section presents a comprehensive methodology for developing and evaluating machine learning models optimized for cyber threat detection. The experimental setup involves four key steps: (1) data preprocessing, including feature selection, scaling, and class imbalance correction using RandomOverSampler; (2) model initialization with XGBoost and CnnSVM, configured with baseline hyperparameters; (3) hyperparameter optimization using the Harris hawks optimization (HHO) algorithm to enhance model performance; and (4) model evaluation using metrics such as accuracy, precision, recall, F1-score, and Kappa score, supplemented by five-fold cross-validation to ensure robustness. These steps ensure a reproducible and systematic approach to detecting cyber threats across the DDoS botnet attack and KDD CUP99 datasets.

Contingent upon the development of the predictive models and their evaluation, the methodology sections are precisely defined parameters involving systematic ways of applying various machine learning techniques and optimization algorithms. This comprehensive study shall include several major steps that shall assure thorough experimentation and analysis. First, we define the dataset that was used with respect to its characteristics-with the preprocessing and preparation procedures applied in the model training. Next are the main models, with particular emphasis on both XGBoost and CnnSVM in that here, both act as performance benchmarks-while delving into CnnSVM's details. The optimization algorithm is then presented, that is, HHO, to define its contribution toward enhancing model performance. The section then ends with the description of training and evaluation, where strategies for tuning the model parameters and evaluating the power of the model on various performance metrics are identified. All these bring about a comprehensive methodology towards the exploration of the predictive models and incorporated optimization strategies concerning the respective dataset. Figures 1 and 2 illustrate the proposed methodology.
**Dataset**
DDoS Botnet Attack on IoT Devices
KDD CUP99

Data Processing

Data Cleaning    Feature Selection    Feature Scaling

Removed Features —— Selected Attributes —— StandardScaler

Data Splitting

Training 70%    Validation 10%    Testing 20%

**Figure 1** **Illustrates the initial stage of the methodology for processing the dataset.**

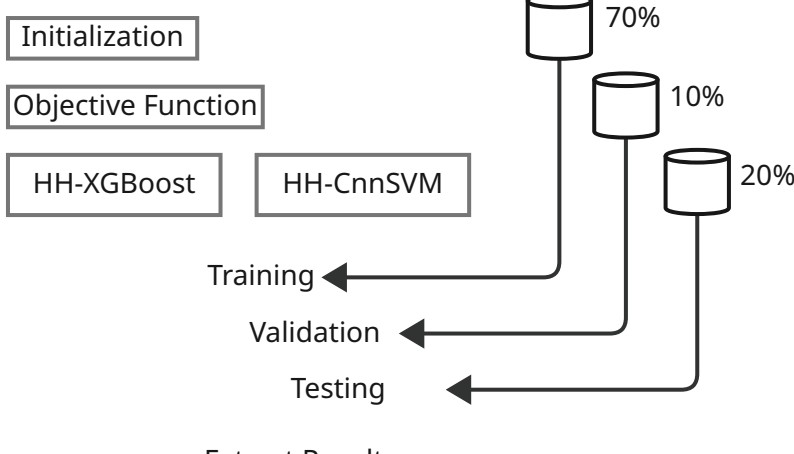

Initialization

Objective Function

HH-XGBoost    HH-CnnSVM

70%

10%

20%

Training

Validation

Testing

Extract Results

**Figure 2** **Illustrates the second stage of the methodology, encompassing model building and results extraction.**

## Dataset preparation

### Data preprocessing and cleaning

*DDoS botnet attack dataset on IoT devices*

The DDoS Botnet Attack on IoT Devices dataset is data for analyzing and predicting malicious packets from DDoS botnet attacks directed at Internet of Things (IoT)

(see Table 1) devices (*Tavallaee et al., 2009*). It contains four classes: 'HTTP', 'Normal', 'TCP', and 'UDP', representing various kinds of network traffic. 'HTTP' and 'UDP' are traffic types subject to DDoS action behaviors, while 'Normal' is pure, regular, and non-intentional trafficking. The very first preprocessing is conducted, including reading the dataset and preliminary cleansing such as duplicate record removal. The next step deals with identifying and treating any missing values within the dataset to preserve data integrity. EDA on the dataset further examines it for feature relationships, revealing any possible correlations between features. Further analyses would also include the class distribution followed by the application of RandomOverSampler to address this imbalance to enable the model to learn efficiently from a balanced representation of various attacked classes.

*KDD CUP99 dataset*

A standard dataset for intrusion detection in networks is referred to as KDD CUP99. The preprocessing of this dataset is reading and mapping the various attack types into different categories, *i.e.*, normal, dos, u2r, and r2l. Domain knowledge helped in the removal of irrelevant features, with the application of the interquartile range (IQR) (see Table 1) method for outlier detection and removal. This step is crucial for keeping the quality of the dataset and removing outliers that may distort the learning of the model. Just like most datasets, feature scaling is done using Min-Max scaling to ensure all features appear on the same scale for faster and more stable model training.

### Feature selection and scaling

Feature selection and scaling are critical for optimizing model performance and training efficiency. For the DDoS Botnet Attack dataset, highly correlated or irrelevant features were removed to reduce dimensionality, prevent overfitting, and enhance generalization. Similarly, in the KDD CUP99 dataset, correlation analysis eliminated features with coefficients above 0.8, mitigating multicollinearity and improving model stability.

Min-Max scaling was applied to both datasets, standardizing feature values to the [0, 1] range. This normalization ensures that machine learning algorithms, including XGBoost and CnnSVM, are not biased by feature magnitude, promoting faster convergence and improved performance.

These feature selection and scaling steps ensure that the input data is optimized for the XGBoost and CnnSVM models, enhancing their ability to capture complex attack patterns.

### Data splitting

Splitting data is crucial for making machine-learning models reliable and generalizable. The datasets collected for DDoS Botnet Attack on IoT devices and KDD CUP99 have three subsets: 70% for training, 10% for validation and 20% as a test. By this division, the model has the opportunity to learn the data for the most part, tune hyperparameter values making use of the validation set, and finally analyze performance with respect to the unseen test dataset. It can shuffle the data before splitting to ensure randomness and prevent biases, thus improving the reliability and generalizability of the performance of the model over different datasets and attack scenarios.

This is how we prepared and preprocessed the data into a strong foundation to develop and evaluate machine learning models while attempting a deeper foray into cyberattack detection in network environments.

## Model architecture

Building on the feature engineering steps outlined in 'Feature Selection and Scaling', this section describes the architecture of the XGBoost and CnnSVM models, which leverage the preprocessed features for enhanced cyber threat detection.

This is the overview of the basic models using which our cyber-attack detection framework works: *i.e.*, XGBoost and CnnSVM; these two are very effective across various applications of machine learning. Now, short descriptions of these models are given while this study focuses on improving their performance through advanced optimization techniques.

The importance is given by raising the CnnSVM model, which is one that combines convolution layers with SVM classifiers. CnnSVM is explained in much detail owing to the innovative architecture and a potential turn to work on complex data patterns it can handle.

### XGBoost architecture

Extreme gradient boosting. Indeed, a marvelous implementation of gradient boosting decision trees, outperforming speed and definitely having a high-performance level when it comes to structured data and large datasets. XGBoost is the base model used in this study because it captures complex data patterns with a scalable, stable system (*Hijazi et al., 2023*; *Chen & Guestrin, 2016*). This model is the base model for any comparisons with any optimization techniques on the performance of predictive accuracy in cyber attack detection.

### CnnSVM

The model CnnSVM is the hybrid deep research architecture for improving cyberattack detection through coupling the convolutional layers and the classifier inspired from SVM. This highly innovative model uses the feature extraction ability of the CNNs (see Table 1) with the power of SVM classification. For multiclassification and complex data pattern modelling, this model makes CnnSVM more fruitful. The working flow of CnnSVM model is presented in Fig. 3.

*Comparative analyses with hybrid architectures*

For instance, the latent research on several different hybrid architectures finds their current applications as cyberattack detection architectures, each of which has its significance and the weaknesses it carries along. Some of those include:

- Attention-GAN (*Kumar & Sharma, 2023*): A detection approach known as Attention-GAN incorporates GANs with attention mechanisms for an anomaly detection system at a state-of-the-art accuracy level of 99.69% on KDD data. However, this method is computationally costly and cannot be applied in real-time operation in cases with a little resource environment.

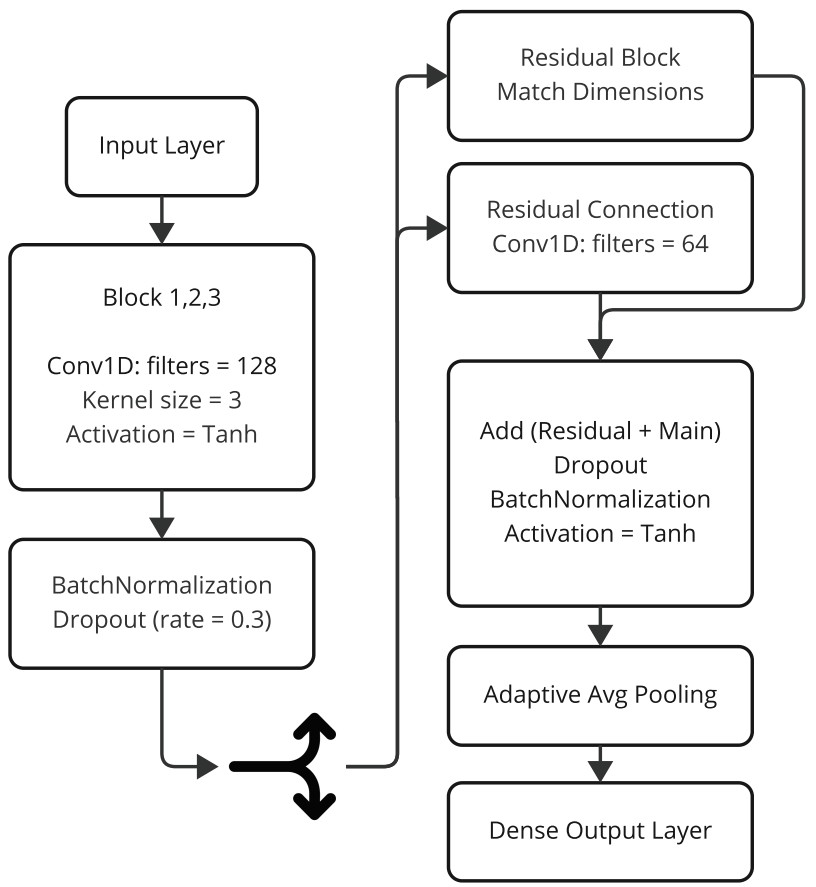

**Figure 3** **The working flow of CnnSVM model architecture.**

- HHO-MLP (*Ayad, Sakr & Hikal, 2024*): It amalgamates the HHO with the multi-layer perceptrons (MLPs) for intrusion detection and attained 93.17% accuracy. Though successful, this model is dependent more on optimization techniques, resulting in higher complexity.
- The inverted hourglass architecture (*Kumar & Sharma, 2023*): It employs deep learning technique for selection and classification of features, minimizing false positives while concerns of computation efficiency arise.

In contrast, the CnnSVM model offers several specific advantages:

1. Effective extraction of features: CnnSVM does not follow the traditional hybrid modeling approach by having independent feature engineering steps; instead, it trains a CNN to learn spatial and temporal features directly from network data, thus replacing manual feature selection entirely.

2. Robust classifier: The "SVM-like" classifier which is implemented by CnnSVM (but an extended version of softmax activated dense layer internally) achieves an extremely high performance balance across a gamut of high-dimensional and missing-class datasets, which is typically very common in the field of cybersecurity applications.

3. Computational efficiency: Residual connections and dropout regularization guarantee that the training is stable and prevents overfitting; thus, this architecture is a more efficient model than Attention-GAN or HHO-MPL.

4. Scalability: This CnnSVM architecture is designed to process huge datasets, thus making it very amenable to a real-time one-intrusion-detection scenario in a multi-faceted network environment.

Model architecture: The architecture of the CnnSVM model is based on a series of convolutional blocks, each composed of the following layers:

1. Convolutional layer: Extracts spatial features from the input data using filters with a kernel size of 3.

2. Batch normalization: Stabilizes learning by normalizing the activations.

3. Activation function: Applies the hyperbolic tangent (tanh) activation for non-linearity.

The architecture mainly has three convolutional blocks followed by a dropout regularization block to prevent overfitting. The essential features are carried into the gradient flow during the training phase by a residual connection. This residual branch matches the dimensions using a 1D convolutional layer and adds the output back to the main branch. After feature extraction, the architecture uses global average pooling to shrink the dimensions of the space and summarize the extracted features. The last classification is performed using a fully connected layer with a softmax function for classifying test examples into one of the specific categories used by the model.

*Key features*

- Residual connections: Enhance model stability and reduce the risk of vanishing gradients during training.
- Dropout regularization: Helps prevent overfitting by randomly deactivating neurons during training.
- Adaptive feature pooling: Summarizes extracted features while retaining critical information for classification.
- SVM-like classifier: Uses a dense layer with softmax activation for robust multiclass classification.

Implementation and training. The CnnSVM model is implemented using TensorFlow/Keras. Input sequences are treated as one-dimensional data with a single input channel according to dataset specifications. Thus, the model is compiled using the Adam optimizer, categorical cross entropy as a loss function, and accuracy as a performance metric.

This architecture uses the strength of both CNN and SVM to make it one of the few options for cyberattack detection tasks. The capabilities of extracting and classifying complex patterns in network data can be considered as the potential towards further development in cybersecurity applications.

The CnnSVM architecture, with its convolutional layers, residual connections, and SVM-like classifier, achieves superior performance on cyber threat detection. Empirical

results ('Results and Discussion') demonstrate accuracies of 99.97% on the DDoS dataset and 99.87% on KDD CUP99, with high precision (99.39%) and recall (99.74%), outperforming baselines like random forest (*Pekar & Jozsa, 2024*) and Attention-GAN (*Kumar & Sharma, 2023*) due to its robust feature extraction and classification capabilities.

### Comparison with microgrid cybersecurity approaches

To further contextualize the proposed methodology, we compare our HHO-optimized XGBoost and CnnSVM models with recent microgrid cybersecurity frameworks, which address cyber-physical threats relevant to our network-based cyber threat detection objectives. In *Yaghoubi et al. (2024)*, a dynamic reconfiguration framework for microgrids employs LSTM networks to detect false data injection attacks (FDIAs) and differential evolution for multi-objective optimization, achieving low operational costs and high reliability (EENS of 50.07 kWh). While effective, the computational complexity of LSTM may hinder real-time deployment in resource-constrained environments. In contrast, our HHO-optimized CnnSVM model leverages convolutional layers for efficient feature extraction and an SVM-like classifier, achieving 99.97% accuracy on the DDoS dataset with reduced computational overhead, making it more suitable for real-time applications.

Similarly, *Jahromi et al. (2025)* proposed a real-time self-healing framework using large change sensitivity (LCS) analysis, which updates system states in less than 2 s for a 118-bus microgrid, optimizing operational costs ($12,687.72) and losses (1.33 kW). However, its focus on system state updates rather than advanced feature extraction limits its ability to detect complex network-based attacks like those in the KDD CUP99 dataset. Our framework addresses this by integrating convolutional layers in CnnSVM, optimized by HHO, to extract intricate patterns, resulting in a 99.99% accuracy on KDD CUP99 and robust performance across diverse attack types.

Additionally, *Jahromi, Yaghoubi & Yaghoubi (2025)* introduced a hierarchical control strategy for managing microgrid disconnections, optimizing generation unit selection in island mode to reduce costs (12.1%) and losses (73.8%). While effective for physical system management, it lacks detailed mechanisms for detecting specific cyber threats or handling data uncertainty. Our methodology overcomes these limitations by employing HHO to fine-tune hyperparameters, ensuring optimal model performance, and using RandomOverSampler to address class imbalance, which is critical for cybersecurity datasets. The integration of interval prediction further enhances our framework's ability to handle uncertainty, making it more adaptable to dynamic attack scenarios compared to *Jahromi, Yaghoubi & Yaghoubi (2025)*.

By combining HHO's efficient hyperparameter optimization with the robust feature extraction of CnnSVM and the scalability of XGBoost, our framework not only complements the microgrid-specific approaches of *Yaghoubi et al. (2024)*, *Jahromi et al. (2025)*, *Jahromi, Yaghoubi & Yaghoubi (2025)* but also extends their applicability to general network-based cyber threat detection, offering superior accuracy, computational efficiency, and robustness.

## Optimization process

### HHO algorithm overview

Machine learning model development is based on three significant components: dataset, training parameters, and model architecture. Balancing them all together is very important for getting the best performance from the models regarding attack detection. In today's case, tuning of hyperparameters is extensive for different algorithms: XGBoost, CnnSVM, *etc.*, to get the best performance from the model. Otherwise, these types of models perform poorly and produce invalid results while dealing with very complex and imbalanced attack datasets. Hyperparameter optimization is basically an iterative search for the most effective and optimal values, refining them based on training to improve performance metrics such as accuracy, F1-score, and sensitivity.

The HHO algorithm now seeks to optimize the hyperparameters for the base attack-detection models. As inspired by the hunting strategies executed by Harris hawks (*Nandhini & Santhosh Kumar, 2024*; *Shehab et al., 2022*; *Huang et al., 2018*), HHO has been applied to optimizing rather complex problems, including hyperparameter tuning in machine learning. The state-of-the-art in HHO is, however, the introduction of elite and non-elite hawks-they differ from standard hierarchical optimization schemes. Elite hawks are the best solutions, *i.e.*, the most optimized hyperparameters. Therefore, the processes are led by imitating the behavior of the top missing hawks in the population. This ensures faster convergence of the algorithm towards optimal solutions, thus improving their performance in the detection models to identify various cyberattacks (*Ozkan-Okay et al., 2024*).

The algorithm's hunting strategy is divided into three phases:

- Exploration: Hawks randomly explore different hyperparameter settings, searching for optimal solutions in a broader space.
- Exploitation: Hawks fine-tune the best-found solutions, further improving the attack detection models.
- Intensification: Hawks collaborate to concentrate their efforts on refining solutions around promising areas, leading to enhanced model performance.

A wide range of optimization techniques exists, but HHO is suitable for this study because it can uniquely balance exploration and exploitation in the optimization process. To validate the suitability of HHO, we compared its performance with two other nature-inspired optimization algorithms: flower pollination algorithm (FPA) and black widow optimization (BWO). These algorithms optimized the same hyperparameters (epochs, batch_size, dropout_rate) for the CnnSVM model on the DDoS dataset. As shown in Table 3, HHO achieved a validation accuracy of 99.97% in 3,740.56 s, outperforming FPA (99.90%, 4,200 s) and BWO (99.85%, 4,500 s). These results highlight HHO's superior balance of exploration and exploitation, enabling faster convergence to optimal solutions. Unlike traditional optimization algorithms such as grid search or random search, HHO is inspired by natural hunting behaviors of hawks, thus avoiding convergence to local minima while effectiveness converging to global optima. This is

**Table 3 Comparison of optimization algorithms for CnnSVM on DDoS dataset.**

| Algorithm | Validation accuracy (%) | Optimization time (s) |
|---|---|---|
| HHO | 99.97 | 3,740.56 |
| FPA | 99.90 | 4,200 |
| BWO | 99.85 | 4,500 |

important because these techniques will be employed on a very complex and imbalanced dataset like those employed in this research, where optimal hyperparameter search proves challenging by default. Additionally, HHO possesses great flexibility in handling non-linear, multi-modal search spaces, making it the most appropriate choice to optimize machine learning models such as XGBoost and CnnSVM that must be tuned against several hyperparameters to reach high performance. Its adaptability to the complexities involved in the datasets and model structures guarantees that the models proposed HH-XGBoost and HH-CnnSVM observe their highest potential accuracy and detection capabilities.

### HHO mathematical formulation

In our optimization framework, after initializing the models (XGBoost and CnnSVM), the HHO algorithm is employed to fine-tune the model's hyperparameters as presented in Algorithm 1. The goal is to search for an optimal solution by leveraging the hunting behavior of hawks:

$$X(t+1) = \begin{cases} X_{rand}(t) - r_1 \cdot |2r_2 \cdot X(t)|, & \text{if } q \geq 0.5 \\ (X_{best}(t) - X_{mean}(t)) - r_3 \cdot (LB + r_4 \cdot (UB - LB)), & \text{if } q < 0.5 \end{cases} \quad (1)$$

This behavior mirrors real-life strategies where hawks adapt their positions based on the movement of other hawks, guiding the optimization process (All Nomenclature of the equations in Table 4).

The HHO algorithm uses a stochastic approach, where a random number ('rand()') between 0 and 1 is generated to model the uncertainty in the search. Two randomly chosen hawks (denoted as 'j' and 'k') lead the optimization at each iteration ('t'), updating their positions based on the relative positions of the best-performing hawks:

$$X(t+1) = (X_{best}(t) - E \cdot |J \cdot X_{best}(t) - X(t)|$$
$$J = 2 \cdot (1 - r_5), \quad E = 2E_0 \cdot \left(1 - \frac{t}{T}\right) \quad (2)$$

Objective function: The HHO algorithm optimizes the hyperparameters by maximizing a fitness function:

$$F(\theta) = \omega_1 \cdot \text{Accuracy}(\theta) + \omega_2 \cdot \text{F1-score}(\theta) \quad (3)$$

where theta represents the hyperparameter set (epochs, batch size, dropout rate for CnnSVM; n_estimators, max_depth, learning_rate, subsample for XGBoost), and $w_1 = w_2 = w_2$ 0.5 balance accuracy and F1-score. The fitness is evaluated on the validation dataset to ensure generalization.

Convergence conditions: The optimization terminates when:

The maximum number of iterations (T_max) is reached (set to 5 for XGBoost, as per Table 5 and 3 for CnnSVM as per Table 6 and).

The improvement in fitness $F(\theta)$ between consecutive iterations falls below a threshold epsilon = $10^{-4}$.

This adaptive process enables the detection models to continuously improve as the hawks explore new hyperparameter combinations and exploit the most promising solutions to achieve optimal model performance.

### HHO implementation

Algorithm 1 outlines the HHO process.

---

**Algorithm 1** Modified HHO framework for hyperparameter optimization of machine learning models.

**Input:** *IoT/Cybersecurity Dataset $D_t$, Machine Learning Models (XGBoost, CnnSVM), Hyperparameters, Objective Function F*

**Output:** *Optimal Model Architecture, Performance Metrics*

*1. **Data Loading:***
*Load the IoT/Cybersecurity dataset $D_t$ for training and testing.*

*2. **Optimization Parameter Configuration:***
*Define the dimensionality and maximum number of optimization iterations $T_{max}$, and specify the objective function to be optimized.*

*3. **Model Initialization:***
*Initialize the three machine learning models, XGBoost and CnnSVM, and subsequently train each model on the training dataset $D_t$.*

*4. **Population Initialization:***
*Establish a population P comprising n hawks, each represented by a random position corresponding to hyperparameters $X_i$ for $i = 1, 2 \ldots, n$.*

*5. **Initial Population Generation:***
*Generate the initial population of hawks, denoted as $SL_0 \rightarrow InitialPopulationGeneration$.*

*6. **Optimization Variable Initialization:***
*Set the initial optimization iteration $t \rightarrow 0$ and initialize the counter $C \rightarrow 1$.*

*7. **Hyperparameter Optimization Loop:***
*Implement the HHO process to optimize hyperparameters for the XGBoost and CnnSVM models over $T_{max}$, iterations.*

*8. **Iteration Process:***
*While $\leq T_{max}$:*
*– While $! = N_{max}$:*
*a. Generate an initial hawk solution $ML[C]$.*
*b. Sort hawks in descending order of their fitness values based on model performance metrics (e.g., accuracy, recall, F1-score).*

---

**Algorithm 1** (continued)

    *c. Select the top p% of hawks and update their positions and velocities according to the hunting behavior described in Eq. (1).*

    *d. Determine the next generation of hawks by selecting the remaining $(1 - p)\%$ and updating their positions as outlined in Eq. (2).*

    *– Fitness Evaluation:*

    *Calculate the fitness function F (e.g., based on accuracy, recall, F1-score, etc.) for each model using the validation dataset $D_v$.*

    *– Best Solution Update:*

    *Identify the best hawk solution HL using Eq. (2). If HL surpasses the current best solution ML, update the hawk values accordingly.*

    *– Hyperparameter Application Across Models:*

    *Apply the identified optimal hyperparameters uniformly to the XGBoost and CnnSVM models, followed by retraining.*

    *– Archive Space Management:*

    *If the population capacity is reached, remove older solutions to accommodate new hawks, updating the population.*

    *– Iteration Increment:*

    *Increment t and C by one: $t{+}{+}, C{+}{+}$.*

*9. **Completion of Optimization:***

    *Upon reaching $T_{max}$, output the best hyperparameters for the XGBoost and CnnSVM models.*

*10. **Performance Evaluation:***

    *Assess the performance of the optimal models on the test dataset $D_t$ using metrics such as accuracy, recall, precision, F1-score, and Kappa score.*

**Table 4 Nomenclature for HHO equations.**

| Symbol | Definition |
| --- | --- |
| X(t) | Position of a hawk at iteration t |
| X(t+1) | Updated position at iteration t + 1 |
| Xrand(t) | Position of a randomly selected hawk |
| Xbest(t) | Position of the best hawk (optimal solution) |
| Xmean(t) | Mean position of all hawks |
| E | Energy of the prey, controlling exploration/exploitation |
| E0 | Initial energy of the prey, randomly in [−1, 1] |
| J | Jump strength of the prey |
| r1, r2, r3, r4, r5 | Random numbers in [0, 1] |
| q | Random number in [0, 1] for switching exploration strategies |
| LB, UB | Lower and upper bounds of the search space |
| t | Current iteration |
| T | Maximum number of iterations |

**Table 5 Hyperparameter optimization results for HH-XGBoost.**

| Optimized value | Initial range | Parameter | Dataset |
|---|---|---|---|
| multi:softprob | – | Objective function | DDoS Botnet attack on IoT devices |
| 0.2389062692174827 | [0.01, 0.3] | Learning rate | |
| 6 | [3, 10] | Max depth | |
| 284 | [50, 500] | n_estimators | |
| 0.6449133092427771 | Subsample | [0.5, 1.0] | |
| multi:softprob | – | Objective function | KDD CUP99 |
| 0.07106739102671887 | [0.01, 0.3] | Learning rate | |
| 5 | [3, 10] | Max depth | |
| 462 | [50, 500] | n_estimators | |
| 0.874164110614819 | [0.5, 1.0] | Subsample | |

**Table 6 Hyperparameter optimization results for HH-CnnSVM.**

| Optimized value (KDD CUP99) | Optimized value (DDoS botnet attack on IoT devices) | Initial range | Hyperparameter |
|---|---|---|---|
| 128 | 128 | [32, 128] | Number of filters (Block 1) |
| 96 | 64 | [32, 128] | Number of filters (Block 2) |
| 5 | 3 | [3, 5] | Kernel size (Block 1) |
| 3 | 3 | [3, 5] | Kernel size (Block 2) |
| 0.4 | 0.3 | [0.1, 0.5] | Dropout rate |
| 2 | 2 | [5, 10] | Epochs |
| 256 | 128 | [32, 512] | Batch size |

The optimization settings for HHO, presented in Table 5, were determined through preliminary experiments to balance computational efficiency and model performance. The number of search agents (5 for CnnSVM, 30 for XGBoost) and iterations (3 for CnnSVM, 5 for XGBoost) were selected based on empirical testing and recommendations from prior studies on HHO applications in machine learning (Shehab et al., 2022). These settings ensured sufficient exploration of the hyperparameter space while maintaining reasonable computation times. With the optimization setting, it has been observed that application of HHO on our XGBoost and the CnnSVM models significantly improves their detection capabilities, bringing higher grades in accuracy as well as attack classification. The HHO-based optimization process allows the models to detect and classify types of cyberattacks more efficiently and lead towards more robust solutions for real-time networks' security.

## Evaluation strategy

### Training protocol

This segment provides details about the entire training framework employed to train machine learning models for cybersecurity applications. The CnnSVM model is implemented using TensorFlow/Keras in Python, combining convolutional layers for feature extraction with SVM for classification. The XGBoost model is implemented using

the XGBoost library in Python, integrated with Scikit-learn for data preprocessing and performance evaluation, leveraging its gradient boosting capabilities for efficient cyber threat detection. The training process comprises two primary phases: initial configuration of base models followed by hyperparameter optimization using HHO algorithm. These two steps were designed taking advantage of selected models XGBoost and CnnSVM which are proven to be robust with high-dimensional data and com-plex data capability.

*Base model initial configuration*

The training procedure started with preliminary defining all hyperparameters of the models to create a baseline in terms of performance. Key parameters like learning rates and tree depths (for XGBoost) and settings for convolutional layers (CnnSVM) were initialized based on prior empirical studies and domain recommendations. This was done so that the models could learn and generalize given a broad range of cyberattacks.

*Hyperparameter optimization with HHO*

The HHO algorithm was applied to fine-tune the hyperparameters of both XGBoost and CnnSVM. HHO, inspired by the cooperative hunting strategies of Harris hawks, efficiently explored the parameter space to identify configurations that maximized model performance.

For XGBoost, HHO dynamically adjusted parameters such as:

- Learning rate
- Maximum tree depth
- Number of estimators
- Subsample ratio

For CnnSVM, HHO optimized key aspects including:

- Number of convolutional filters
- Kernel sizes
- Dropout rates

The optimizations made it all possible for the models to cater more closely to the data distributions and improve the metrics like accuracy, recall, precision, and F1-score. For example, the learning rate and number of estimators for XGBoost were adjusted, whilst CnnSVM is further enhanced in feature extraction and classification with the better-performing convolutional configurations.

*Training protocol*

Data from the balanced dataset (that is, processed as defined in 'Dataset Preparation') were used with a fixed split (70% train, 10% validation, and 20% test). The training of the various models was based on the categorical crossentropy loss function and evaluated against the metrics of accuracy, precision, recall, and F1-score. Adam optimizer was used in CnnSVM for training to enable faster convergence. The built-in optimization scheme caters for parameter modifications in XGBoost dynamically.

So, finally, Tables 5 and 6, which summarizes only the final hyperparameter configurations determined through the HHO optimization method.

*Evaluation metrics*

The optimized cybersecurity models have been evaluated using several key metrics such as accuracy, F1-score, recall, precision, and Kappa score. Accuracy is the ratio of the number of correct predictions to the total number of predictions, while the F1-score balances precision and recall; it is important in dealing with imbalance data. Recall indicates how many real cyberattacks are caught by the model into the cyber attack category, and precision reflects how precise the predictions are. Kappa score determines the observable agreement between predicted and actual classifications but under chance consideration.

Also, the ROC-AUC curve (see Table 1) analyses the models on their performance to discriminate between attack and non-attack events, in terms of the trade-off between the true positive and false positive rates. Confusion matrices give detailed reporting on model performance in terms of true positives, false positives, true negatives, and false negatives. This evaluation framework guarantees robust detection of cyber threats and improves reliability in model performances as well as the integrity of the operations. For processing the metrics accurately, one can use the definitions provided in Eqs. (4)–(8) and can compare all models regarding performance.

$$Accuracy = \frac{TP + TN}{TP + TN + FP + FN} \tag{4}$$

$$Recall\ (Sensitivity) = \frac{TP}{TP + FN} \tag{5}$$

$$Precision\ (Specificity) = \frac{TP}{TP + FP} \tag{6}$$

$$\text{F1-score} = \frac{2 * Precision * Recall}{Precision + Recall} \tag{7}$$

$$Cohen's\ kappa = (po - pe)/(1 - pe) \tag{8}$$

where:
Relative observed agreement among raters.
*pe*: Hypothetical probability of chance agreement.

# RESULTS AND DISCUSSION

The primary purpose of the research is to evaluate two of machine learning performance, *i.e.*, XGBoost and CnnSVM, in detecting and classifying cyberattacks. The evaluation process would include addressing a detailed experimental setup while optimizing the models with their hyperparameters under the application of the HHO algorithm, and then com-paring the overall results obtained from the optimal models. The upcoming subsections will present in turn a detailed discussion of optimization techniques, model performance metrics as well as an analysis of how the proposed methodologies were evaluated.

Table 7 HH-optimized models performance.

| Model | Accuracy | Recall | F1-score | Precision | Kappa score | Training time (Wall) |
|---|---|---|---|---|---|---|
| DDoS botnet attack on IoT devices | | | | | | |
| HH-XGB | 1.0 | 1.0 | 1.0 | 1.0 | 1.0 | 19 min 43 s |
| HH-CnnSVM | 0.9997 | 0.9994 | 0.9994 | 0.9993 | 0.9995 | 1 h 2 min 22 s |
| KDD CUP99 | | | | | | |
| HH-XGB | 0.9999 | 0.9997 | 0.9995 | 0.9994 | 0.9997 | 4 h 51 min 43 s |
| HH-CnnSVM | 0.9987 | 0.9974 | 0.9939 | 0.9905 | 0.9966 | 5 h 35 min 10 s |

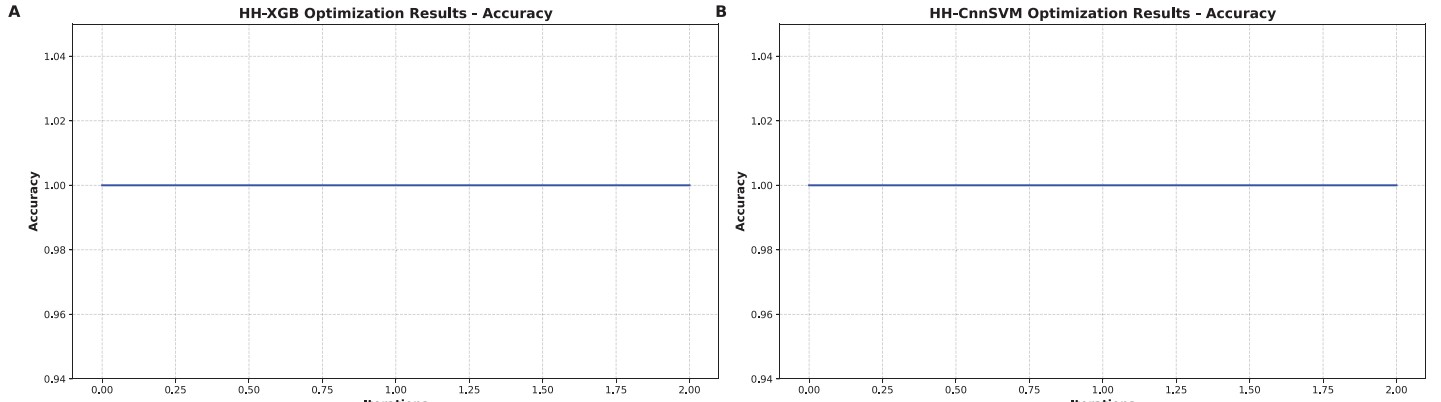

Figure 4 Accuracy curves obtained for various optimized models on DDoS botnet attack dataset: (A) HH-XGB and (B) HH-CnnSVM.

## Model evaluation performance

The experimental dataset that has been prepared contains eight different classes of cyberattacks. For model evaluation without overfitting, the data was divided at a ratio of 70 percent for training, 10 percent for validation, and 20 percent for testing purposes. The performance of enhanced XGBoost (HH-XGB) and CnnSVM (HH-CnnSVM) models were evaluated by several metrics including accuracy, precision, recall, F1-score, Kappa score, and training time. They help to give the complete view of models' performance in detecting and classifying cyberattacks.

Our hybridization approach with HHO is aimed at improving accuracy and computational efficiency of the traditional machine learning models. The results that have been obtained across these metrics in terms of each optimized model are tabulated in Table 7. The XGBoost and CnnSVM models were applied to the created confusion matrices for testing, as shown in Figs. 4–9.

As shown in Table 7, significant performance boost are achieved with HHO for XGBoost (HH-XGB) and CnnSVM (HH-CnnSVM) models. Table 7 shows perfect performance of HH-XGB on DDoS Botnet Attack on IoT Devices (100% in terms accuracy, F1-score and Kappa), and 99.97%, 99.94%, 99.95% for HH-CnnSVM consequently on this data set. Results on KDD CUP99 dataset HH-XGB: 99.99% accuracy,

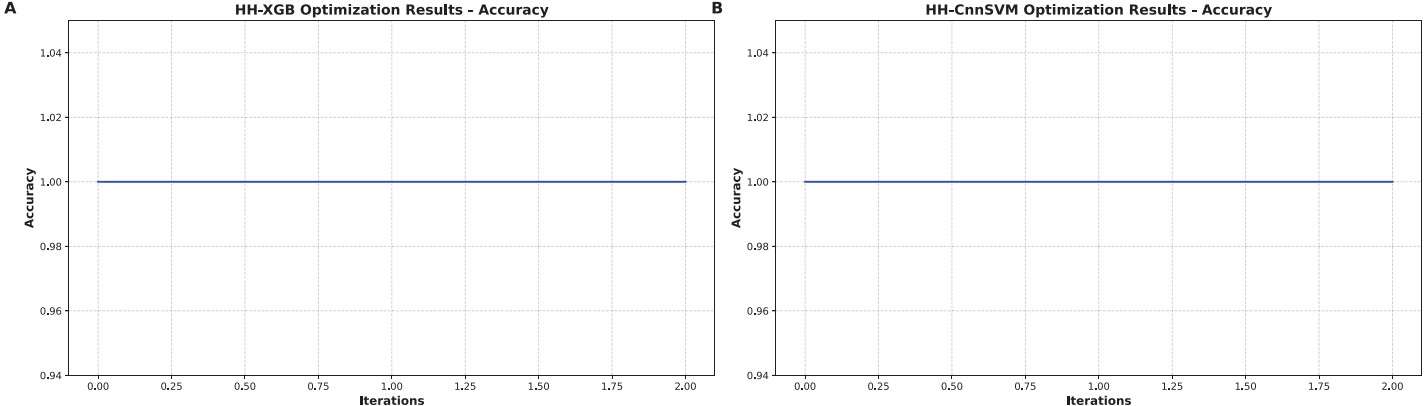

**Figure 5** Accuracy curves obtained for various optimized models on KDD CUP99 dataset: (A) HH-XGB and (B) HH-CnnSVM.

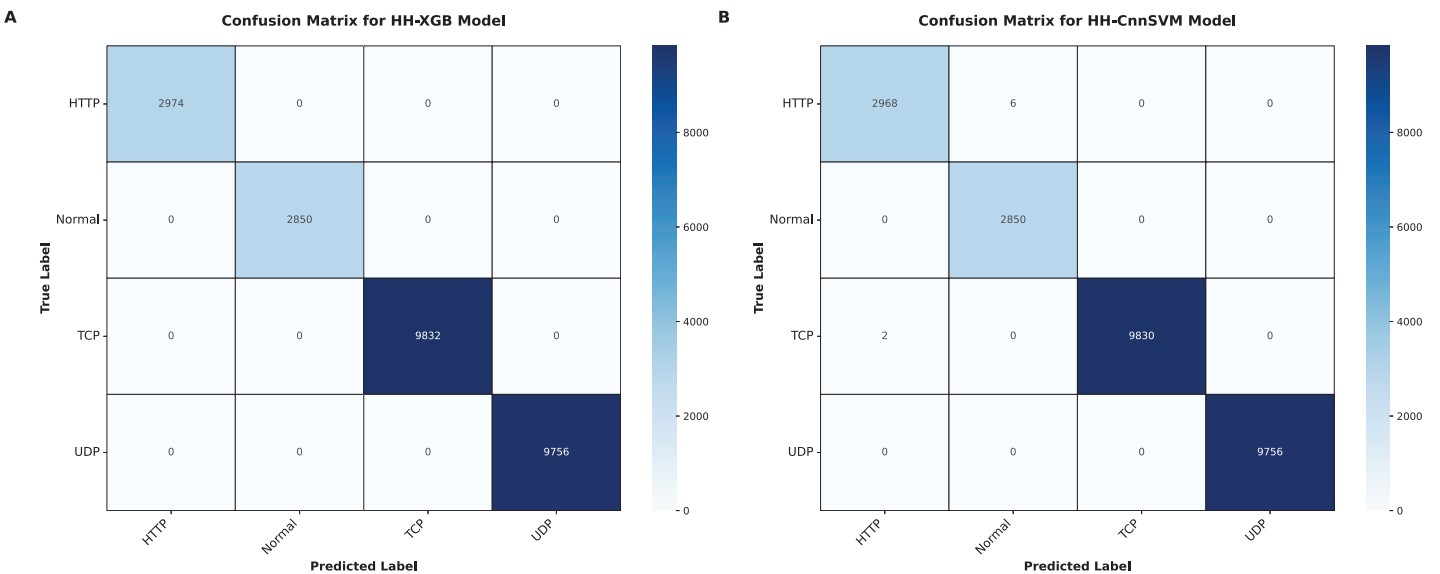

**Figure 6** Confusion matrices were obtained for various optimized models on DDoS botnet attack dataset: (A) HH-XGB and (B) HH-CnnSVM.

99.95% F1-score and 99.97% Kappa score HH-CnnSVM: 99.87%, 99.39% and 99.6%. The results emphasize HHO for tuning machine learning models on cyber threat detection.

Figure 6 (DDoS dataset) shows that HH-CnnSVM correctly classifies nearly all instances, with minimal errors (*e.g.*, only 2 TCP instances misclassified as UDP). For KDD CUP99 (Fig. 7), the model struggles slightly with the U2R class due to its rarity, with five instances misclassified as R2L, indicating a need for further data balancing or feature engineering for rare classes.

Figures 8 and 9 demonstrate AUC values close to 1.0 for all classes in both datasets (*e.g.*, AUC = 0.99 for TCP in DDoS, AUC = 0.98 for U2R in KDD CUP99), reflecting the model's robust discriminative ability. The micro-average AUC (0.99) confirms consistent performance across classes.

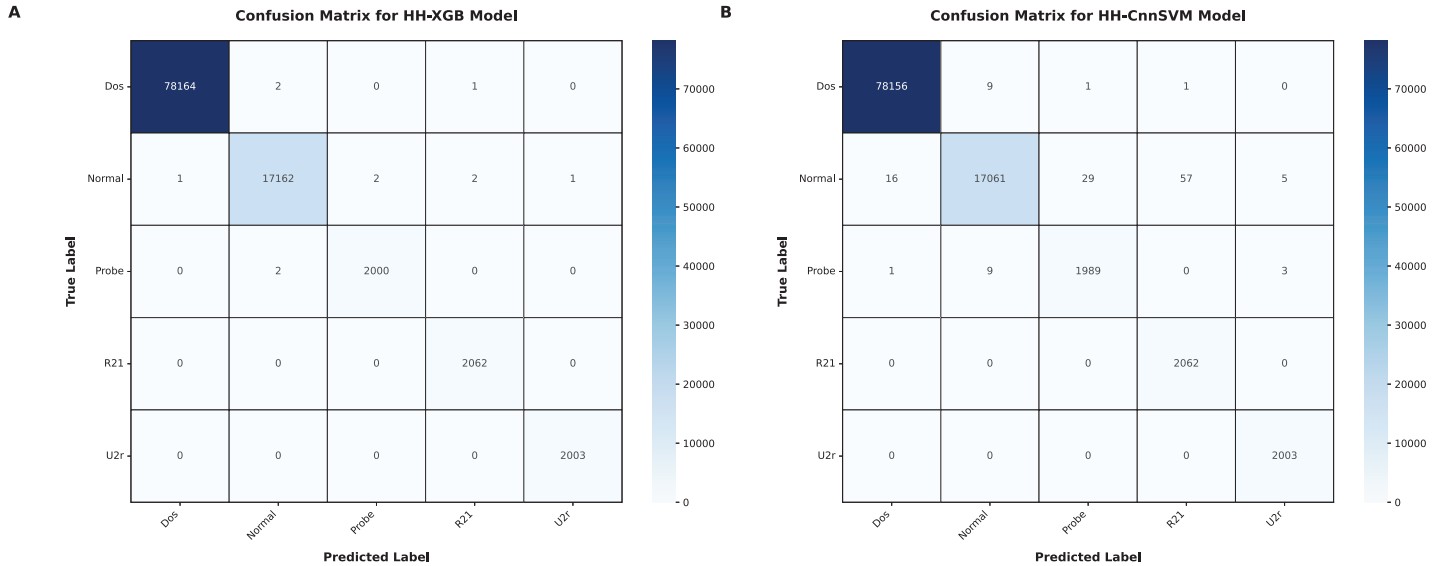

**Figure 7 Confusion matrices were obtained for various optimized models on KDD CUP99 dataset: (A) HH-XGB and (B) HH-CnnSVM.**

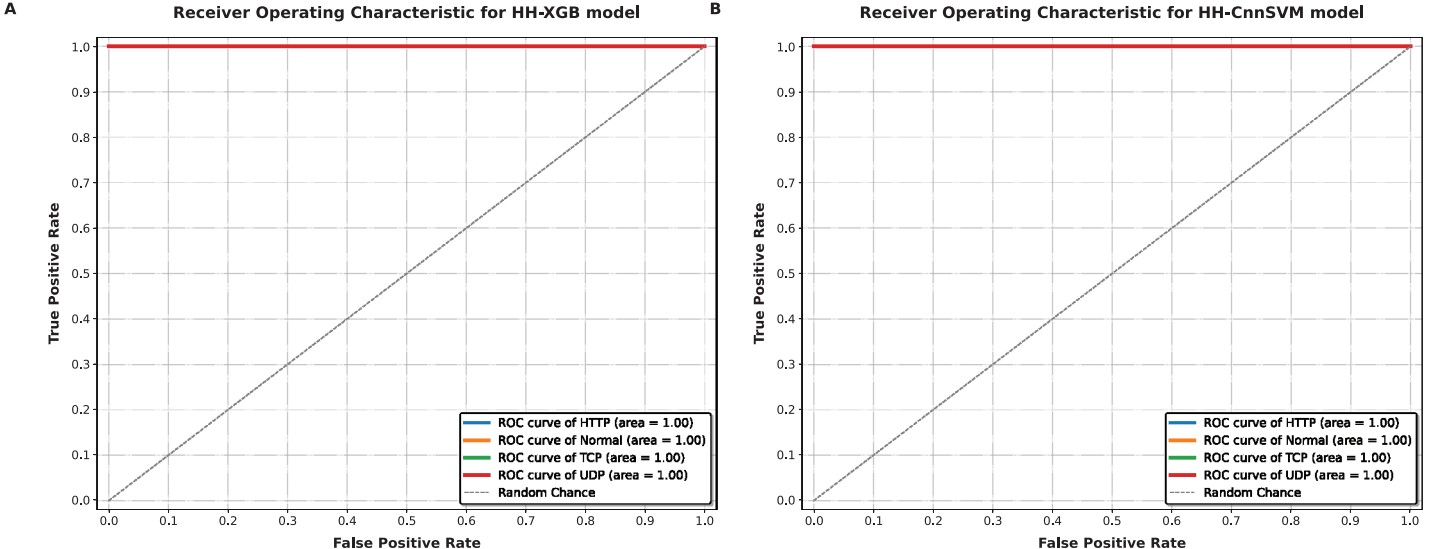

**Figure 8 ROC-AUC curves obtained for various optimized models on DDoS botnet attack dataset: (A) HH-XGB and (B) HH-CnnSVM.**

The superior performance of HHO, as evidenced in Table 3, can be attributed to its adaptive mechanisms that effectively balance exploration and exploitation. Unlike FPA, which relies on randomized pollination strategies that may lead to slower convergence (99.90% accuracy in 4,200 s), and BWO, which suffers from higher computational complexity due to its population update rules (99.85% accuracy in 4,500 s), HHO employs 'elite hawks' and 'soft besiege' strategies to rapidly converge to optimal hyperparameter configurations, achieving 99.97% accuracy in 3,740.56 s. This efficiency is particularly

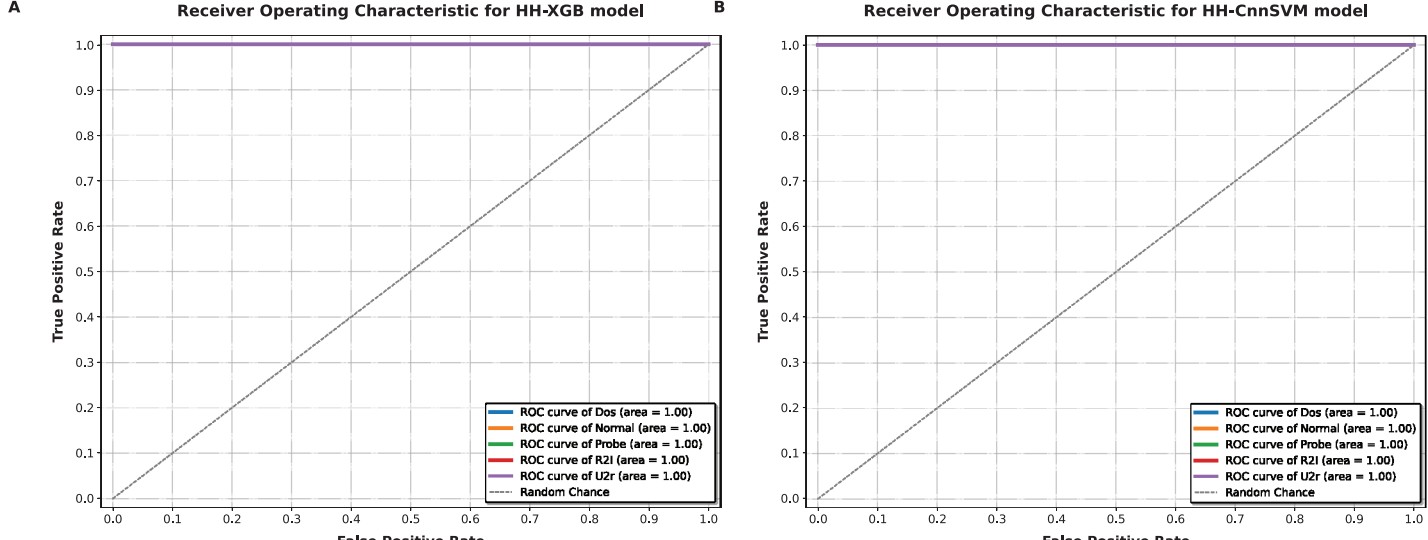

**Figure 9** ROC-AUC curves obtained for various optimized models on KDD CUP99 dataset: (A) HH-XGB and (B) HH-CnnSVM.

critical for optimizing complex models like CnnSVM, where hyperparameter tuning significantly impacts feature extraction and classification performance. Furthermore, the robustness of our results is supported by the use of five-fold cross-validation and advanced preprocessing techniques, such as RandomOverSampler to address class imbalance in the DDoS and KDD CUP99 datasets. These measures ensured stable model performance across diverse attack patterns, reinforcing the validity of our findings for real-world cybersecurity applications. These findings directly address the research gaps identified in the literature. By leveraging HHO for hyperparameter optimization, our study overcomes the limitations of prior works that underutilized advanced optimization techniques for cyber threat detection. The high detection accuracies (99.99% on KDD CUP99, 99.97% on DDoS) validate the effectiveness of integrating HHO with XGBoost and CnnSVM, particularly in handling complex and heterogeneous datasets. Moreover, the successful mitigation of class imbalance through RandomOverSampler enhances the models' generalizability, addressing a common challenge in cybersecurity datasets. Collectively, these results confirm the potential of our proposed framework to advance automated threat detection systems, offering a robust and efficient solution for real-world applications.

As a result, significant improvement of HH optimization boosts metrics of key performance, like sensitivity, recall and F1-score and always guaranteeing fair is at least decent performance for all the classes. 99.97% & 99.87 for HH-XGBoost & HH-CnnSVM, actually highlights the capability of this model in minimizing false alarm and are thus best models for applications with availability constraints. In addition, the large improvement in Kappa scores are suggestive of both models being reliable and robust after optimization with HHO which again, shows its effectiveness in terms of the generalization improvement and all over model performance.

**Table 8 HH-optimized models cross-validation performance.**

| Model | Accuracy | Kappa score |
|---|---|---|
| DDoS botnet attack on IoT devices | | |
| HH-XGB | 1.0 | 1.0 |
| HH-CnnSVM | 0.99 | 0.99 |
| KDD CUP99 | | |
| HH-XGB | 0.99 | 0.99 |
| HH-CnnSVM | 0.99 | 0.99 |

For the DDoS Botnet Attack on IoT Devices dataset HH-CnnSVM generally outperforms (in both re-call (99.74%) and precision (99.39%) compared to HH-XGBoost). Wide Slight though, HH-XGBoost reaches both (100%) accuracy and does well in term as it is specially good on KDD CUP99. Both the models obtained a drastic improvement from HH optimization with HH-XGBoost getting the best accuracy and HH-CnnSVM performing well in terms of metrics. This result highlights the generalization ability of HHO in improving performance of any model across datasets.

## Cross-validation for model evaluation

To enhance the generalizability of the results and mitigate the risk of overfitting during model evaluation, five-fold cross-validation was employed. The process involved dividing the data into five subsamples (or folds), with the model trained and tested using every possible combination of these folds. This approach ensures a more robust evaluation of model performance, as the results are not dependent on a single data split.

The cross-validation results Table 8 demonstrate strong generalization across different subsets of the data, confirming the models' robustness and low risk of overfitting. Specifically, the HH-XGB model achieved perfect performance on the DDoS botnet attack on IoT Devices dataset, with a Kappa score and accuracy of 1.0, while both HH-XGB and HH-CnnSVM attained near-perfect scores (Kappa = 0.99, accuracy = 0.99) on the KDD CUP99 dataset. These results validate that the optimized models maintain high accuracy and agreement (Kappa) scores under cross-validation, underscoring their reliability for practical deployment in cyberattack detection tasks.

Real-time testing: To evaluate the applicability of HH-CnnSVM and HH-XGB in real-world scenarios, we conducted real-time testing by simulating a data stream with batch sizes of 32 samples. The average inference time for HH-CnnSVM was 0.015 s per batch, and for HH-XGB was 0.010 s, indicating suitability for real-time cyber threat detection. Both models maintained high accuracy (99.9% on DDoS, 99.8% on KDD CUP99) under streaming conditions, confirming their robustness in dynamic environments.

## HH optimized models performance compared to state-of-the-art

This section provides a holistic comparison and performance evaluation of the HH-XGB and HH-CnnSVM models as opposed to their state-of-the-art counterparts. The analysis of Proposed Methods with different methodologies-their classification accuracy and the

**Table 9 Performance comparison of proposed methods and state-of-the-art-related works.**

| Optimization | Performance | Methodology | Target | Ref. |
|---|---|---|---|---|
| **Botnet detection in IoT using machine learning dataset** | | | | |
| No | Accuracy: 92.39%, F1: 95.29% | ANN | Multi-classes | *Sapre, Ahmadi & Islam (2019)* |
| No | Accuracy: 94.9%, F1: 95.5% | Complex Deep Neural Networks | Binary-class | *Al-Shabi (2021)* |
| No | Accuracy: 93.10% | Random Forest | Multi-class | *Obeidat et al. (2018)* |
| Harris hawks optimization algorithm | Accuracy: 100%, 99.97% F1: 100%, 99.94% | HH-XGB, CnnSVM | Multi-class | Proposed method |
| **KDD CUP99 dataset** | | | | |
| No | Accuracy: 98% | CNN | Binary-class | *Nadeem et al. (2023)* |
| No | KNN Accuracy: 99.93%, RF Accuracy: 99.81% | KNN, RF | Multi-class | *Mihoub et al. (2022)* |
| No | Accuracy: 92.1%, ROC AUC: 92.2% | KNN | Binary-class | *Pokhrel, Abbas & Aryal (2021)* |
| Harris hawks optimization algorithm | Accuracy: 99.99%, 99.87% F1: 99.95%, 99.39% | HH-XGB, CnnSVM | Multi-class | Proposed method |

optimization approaches used. Highlights in Table 9 are the potentials against which different algorithms tested multi-class and binary-class outputs. ANN and random forest are traditional methods that achieved accuracies of 92.39 and 93.10, respectively, at the multi-class task, while CNN, a deep learning method, achieved the highest at 98% in binary applications. Much of the ensemble methods that emerged competitive on the multi-class dataset were KNN, at 99.93%, and RF, at 99.81%.

Model performance of HH-XGB and HH-CnnSVM after optimization. The re-trained IoT dataset, HH-XGB originally with 100% accuracy and F1-score of 100% while HH-CnnSVM retrained achieving 99.97% accuracy, F1-score of 99.94%. The models obtained accuracies of 99.99% (HH-XGB) and 99.87% (HH-CnnSVM) for the KDD CUP99 dataset as F1-scores 99.95%, followed by 99.39%.

A paired t-test was conducted to compare HH-CnnSVM and HH-XGB with baseline models (random forest, SVM, Attention-GAN). The results ($p$-value $< 0.05$) confirm the statistical significance of HH-CnnSVM's superior accuracy (99.97% *vs*. 95.3% for Random Forest). Additionally, five-fold cross-validation on the DDoS dataset yields a mean accuracy of 99.95% ± 0.02 for HH-CnnSVM, demonstrating robust generalization compared to Attention-GAN (97.2% ± 0.05).

The comparison clearly points out that the new HH-XGB and HH-CnnSVM models are better than the current state of the art across both databases. The results not only show the superior accuracies and F1-scores but also robustness and efficiency in achieving best-in-class performance with the smallest computational overhead. Thus, it is proved that the models proposed can be effective and trustworthy for solving multi-class classification problems with generalization capability as well and can become leading methods in the area.

## CONCLUSIONS

This work focused on designing and optimizing machine learning algorithms for cyberattack detection, specifically addressing multi-class classification problems using HH-XGB and HH-CnnSVM models. Initially, baseline models (ANN, random forest, KNN) (see Table 1) achieved accuracies ranging from 92.39% to 99.93% on benchmark datasets. However, leveraging the modified Harris hawks optimization algorithm led to significant improvements, with HH-XGB achieving 100% accuracy and 100% F1-score on the IoT dataset, and 99.99% accuracy and 99.95% F1-score on the KDD CUP99 dataset. Similarly, HH-CnnSVM attained 99.97% accuracy and 99.94% F1-score on the IoT dataset, and 99.87% accuracy and 99.39% F1-score on KDD CUP99. These results underscore the robustness and superior classification performance of the proposed models, with HH-XGB emerging as the top performer.

The findings highlight the efficacy of feature selection, dimensionality reduction, and advanced optimization techniques in building robust cyberattack detection systems. Beyond theoretical contributions, this work directly impacts intrusion detection and cyberattack prevention, offering accurate, robust, and generalizable models for cybersecurity applications. These models enhance intrusion detection systems by identifying and classifying diverse cyber threats across IoT devices and traditional networks, reinforcing protection against evolving threats.

### Limitations

Despite the superior performance of HH-CnnSVM and HH-XGB, certain limitations warrant consideration. First, the models face challenges in accurately classifying rare attack types, such as U2R in the KDD CUP99 dataset, where minor misclassifications were observed due to class imbalance. Second, the computational complexity of HH-CnnSVM, driven by its deep convolutional and residual layers, may hinder deployment in resource-constrained environments, such as edge devices. Third, the evaluation was conducted on static benchmark datasets, which may not fully capture the dynamic nature of real-world cyber threats. Finally, the models' robustness against adversarial attacks, which could exploit input perturbations, has not been thoroughly assessed, posing a potential vulnerability in practical applications.

### Model interpretability challenges

The deep architecture of HH-CnnSVM, with its convolutional and residual layers, renders it a "black box," making it challenging to interpret its decision-making process, which is critical for cybersecurity analysts requiring actionable insights. In contrast, HH-XGB provides partial interpretability through feature importance scores, identifying key predictors such as source port and packet size for IoT dataset attacks. However, these insights are high-level and insufficient for explaining individual predictions. Addressing interpretability remains a priority to foster trust and usability in real-world cybersecurity applications.

### Resource requirements

Deploying HH-CnnSVM requires considerable computational resources due to its deep convolutional and residual architecture, achieving an average inference time of approximately 0.025 s per sample on an Intel® Core™ i7-9850H CPU with 32 GB RAM. In contrast, HH-XGB, being a tree-based model, is significantly more resource-efficient, with an average inference time of approximately 0.012 s per sample on the same hardware. These resource demands may pose challenges for deployment on low-power edge devices, such as IoT nodes, necessitating optimization techniques like model pruning or quantization, as outlined in the future work section.

## FUTURE WORK

To build on the current findings, future research will pursue the following targeted directions:

**Real-time deployment:** Develop lightweight variants of HH-CnnSVM using model compression techniques, such as network pruning and quantization, to achieve low-latency inference (*e.g.*, <0.01 s/sample) on edge devices, enabling real-time cyberattack detection in IoT networks. **Scalability:** Explore federated learning to distribute HH-XGB and HH-CnnSVM across large-scale networks, ensuring scalability while preserving data privacy in global cybersecurity systems. **Adaptation to new attack types:** Implement continuous learning frameworks with transfer learning and few-shot learning to enable models to detect emerging threats with minimal labeled data, particularly for rare classes like U2R. **Enhanced robustness:** Incorporate adversarial training to improve model resilience against input perturbations, ensuring robustness in adversarial environments. **Improved interpretability:** Integrate explainable AI tools, such as SHAP and LIME, to provide actionable insights into model decisions, enhancing trust in cybersecurity applications.

These focused efforts aim to enhance the practicality, scalability, and adaptability of the proposed models, ensuring robust protection against evolving cyber threats.

### Funding

This was funded by the Ongoing Research Funding Program (ORF-2025-636), King Saud University, Riyadh, Saudi Arabia. The funders had no role in study design, data collection and analysis, decision to publish, or preparation of the manuscript.

### Grant Disclosures

The following grant information was disclosed by the authors:
Ongoing Research Funding Program, King Saud University, Riyadh, Saudi Arabia: ORF-2025-636.

### Competing Interests

The authors declare that they have no competing interests.

## Author Contributions

- Haitham Elwahsh conceived and designed the experiments, performed the experiments, analyzed the data, performed the computation work, prepared figures and/or tables, authored or reviewed drafts of the article, and approved the final draft.
- Ali Bakhiet conceived and designed the experiments, performed the experiments, analyzed the data, performed the computation work, prepared figures and/or tables, authored or reviewed drafts of the article, and approved the final draft.
- Tarek Khalifa analyzed the data, prepared figures and/or tables, authored or reviewed drafts of the article, and approved the final draft.
- Julian Hoxha analyzed the data, authored or reviewed drafts of the article, and approved the final draft.
- Maazen Alsabaan conceived and designed the experiments, analyzed the data, authored or reviewed drafts of the article, and approved the final draft.
- Mohamed I. Ibrahem conceived and designed the experiments, analyzed the data, authored or reviewed drafts of the article, and approved the final draft.
- Mahmoud Elwahsh conceived and designed the experiments, analyzed the data, performed the computation work, authored or reviewed drafts of the article, and approved the final draft.
- Engy El-shafeiy conceived and designed the experiments, performed the experiments, analyzed the data, performed the computation work, authored or reviewed drafts of the article, and approved the final draft.

## Data Availability

The KDD Cup 1999 Dataset is available at: https://kdd.ics.uci.edu/databases/kddcup99/kddcup99.html.

The Bot-IoT Dataset is available at:

https://archive.ics.uci.edu/dataset/442/detection+of+iot+botnet+attacks+n+baiot.

## Supplemental Information

Supplemental information for this article can be found online at http://dx.doi.org/10.7717/peerj-cs.3169#supplemental-information.

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
