# Peer review of "Hyperparameter optimization of XGBoost and hybrid CnnSVM for cyber threat detection using modified Harris hawks algorithm"

_PeerJ Computer Science, doi:10.7717/peerj-cs.3169_

## Round 0.1 · original submission · Major Revisions

Reviewer 1 ·

Basic reporting

The article lacks clarity in several areas, particularly in the abstract and introduction. The statement in the abstract regarding HHO’s ability to enhance machine learning performance and cybersecurity is too general and should briefly explain the specific strategy used to achieve this improvement. Additionally, the introduction contains improperly cited references, such as references [2] and [3], which either do not match the cited sources or do not relate to the mentioned techniques (XGBoost and CNN-SVM). The related work section is weak and lacks a clear identification of research gaps. Furthermore, there are several typographical errors throughout the manuscript, such as in lines 134, 156, 172, and 368, which should be corrected to enhance readability.

Experimental design

The methodology is poorly represented, making it difficult to fully assess the experimental setup. The claim that HHO is suitable due to its balance of exploration and exploitation (Line 386) is not well justified. To support this, the authors should conduct comparative experiments with other optimization techniques such as the Flower Pollination Algorithm and Black Widow Optimization. Additionally, the optimization settings in Table 2 (Line 467) require further clarification regarding the selection criteria for the chosen values. Without this information, it is unclear whether these parameters were empirically determined, based on previous studies, or arbitrarily chosen.

Validity of the findings

The results and discussion section is poorly represented and does not sufficiently support the study's claims. The authors should improve this section by providing a more in-depth analysis of their findings, including a discussion on why HHO outperforms or differs from other optimization algorithms. Moreover, the identified contributions in lines 95 and 98 should be reconsidered, as points 3 and 4 do not reflect novel contributions by the authors. A more robust discussion is needed to establish the validity of the findings.

Additional comments

The overall structure of the paper needs improvement, particularly in presenting the methodology and results. The related work section does not effectively highlight research gaps, making it difficult to understand how this study contributes to the field. Additionally, the presentation of the methodology should be revised to provide a clearer and more structured explanation of the proposed solution. Finally, the authors should conduct a thorough proofreading to eliminate typographical and formatting errors.

·

Basic reporting

Thank you for your submission and the extensive efforts evident in your work. The topic is very relevant and the experiments and findings are encouraging. However, in order to further improve the quality and clarity of the manuscript, I have the following constructive suggestions:

Experimental design

8. The research contribution is somewhat incomplete. It is useful to explicitly highlight the novelty of the work, especially the advantages of using Harris Hawkes Optimization (HHO) compared to previous optimization methods. Provide a dedicated "Contributions" subsection clearly.

9. It is mentioned that CnnSVM is implemented in TensorFlow/Keras, no information is provided about the environment or tools for XGBoost or the HHO algorithm. Please mention these (like Python, Scikit-learn, MATLAB, etc.).

Validity of the findings

10. Although the tables, benchmarks, and experimental results are informative, the explanations provided for them are scattered. It is suggested that the flow and organization of the text be improved so that the reader can better understand and follow the results.

Additional comments

1. Figures 1-9 are not clear and blurry. Please replace them with higher resolution figures.
2. Use the abbreviations table.
3. Figure 3 is difficult to read due to its vertical layout. It is recommended that you redraw it in horizontal format for better readability. You can also delete the current version entirely and consider a newly designed diagram.
4. On line 368, the phrase "Harris Hawkes Optimization (hho)" should be written in uniform capital letters: "Harris Hawkes Optimization (HHO)".
5. You can review and cite the following related articles to strengthen your literature background and methodological position. . These articles are relevant to your work:https://www.mdpi.com/2227-7080/12/10/197,https://www.mdpi.com/1996-1073/18/1/190, https://link.springer.com/article/10.1007/s00202-025-03036-4.

6. The ""Proposed Method Solution" is long, repetitive, and sometimes lacks structural clarity. Consider dividing it into subsections such as:
Dataset preparation, model architecture, optimization process, and evaluation strategy.

Equations (1) and (2) describing the HHO algorithm are not clearly typed and lack parameter definitions. Please correct them for clarity and proper formatting.
7. Some parts of the text are repetitive or written informally, i.e., phrases such as "as shown in", "this shows, "which our cyber-attack detection framework or “We discovered additional shortcomings" should be rewritten in a more formal academic style. The manuscript should follow the dominant tone and structure of the paper.

·

Basic reporting

Authors presented an Optimized XGBoost and Novel Deep Learning Models for Cyber Threat Detection Across Diverse Datasets Using Modified Harris Hawks Optimization. However paper needs extensive revision in the following areas in order to ensure alignment with Peerj computer science standards.

1.Title should be more concise while retaining impact, suggesting to: “Hyperparameter Optimization of XGBoost and Hybrid CnnSVM for Cyber Threat Detection Using Modified Harris Hawks Algorithm”
2.Abstract has unclear separation of background, methods, results, and conclusion which needs more clear components to highlight the core backgrounds, motivations, gaps, and the contributed methodology. The Introduction section Lacks logical flow from problem to methodology. Contributions not highlighted or clearly argued.Need to revise.

3.The research gaps are not clearly discussed in Related Work section must summerized with critical synthesis. Needs comparison of methods and identification of gaps. Author can add a paragraph comparing existing techniques vs. your approach so that reader can get a clear flow to read. It is mendatory to include a table comparing state of the art methods in terms of datasets, accuracy, complexity, and novelty.

4.In methodology section, some minor issues such as: Subsections need clearer separation and numbering. Can improve to section numbering such as 3.1 → 3.1.1, 3.1.2. Clarify distinctions between models and feature engineering. Simplify repetitive statements, especially around feature selection and scaling.

5.The CnnSVM Section should present a clear diagram with the contributed highlights with an empirical evidence based analysis. Present Algorithm 1 as a formal pseudocode block with numbered steps. Use mathematical notations for objective functions and convergence conditions.

6.In result and discussion section the Tables/figures are referenced but not discussed in-depth.Need extended analysis on the figures/tables into the discussion. Compare with baseline models and SOTA more critically (e.g., use paired t-test or cross-validation results). Provide detailed explanation of confusion matrices and ROC curves. Real-time testing is required.
7.The Conclusion and future work needs more reflection on limitations. Future work could be more focused. Must highlight the model interpretability challenges. Resource requirements (especially for deployment).

8.Language & Formatting: Perform full language polishing for grammar, clarity, conciseness, and academic tone. Overuse of passive voice and verbose expressions. Follow PeerJ formatting style (headings, figure captions, references).

Experimental design

In methodology section, some minor issues such as: Subsections need clearer separation and numbering. Can improve to section numbering such as 3.1 → 3.1.1, 3.1.2. Clarify distinctions between models and feature engineering. Simplify repetitive statements, especially around feature selection and scaling. The CnnSVM Section should present a clear diagram with the contributed highlights with an empirical evidence based analysis. Present Algorithm 1 as a formal pseudocode block with numbered steps. Use mathematical notations for objective functions and convergence conditions.

Validity of the findings

In result and discussion section the Tables/figures are referenced but not discussed in-depth.Need extended analysis on the figures/tables into the discussion. Compare with baseline models and SOTA more critically (e.g., use paired t-test or cross-validation results). Provide detailed explanation of confusion matrices and ROC curves. Real-time testing is required.

---

## Round 0.2 · accepted · Accept

The reviewers are satisfied with the recent changes, and therefore, I can recommend this article for acceptance.

Reviewer 1 ·

Basic reporting

The authors perform the requested comments.
No more comments.

Experimental design

-

Validity of the findings

-

·

Basic reporting

I have carefully reviewed the manuscript once again, and considering the revisions made, I approve it.
At this stage, I have no further questions or concerns. Thank you for your efforts.

Experimental design

-

Validity of the findings

-